# Morphology and Properties of Polylactic Acid Composites with Butenediol Vinyl Alcohol Copolymer Formed by Melt Blending

**DOI:** 10.3390/molecules28083627

**Published:** 2023-04-21

**Authors:** Jian Xing, Rongrong Wang, Shaoyang Sun, Ying Shen, Botao Liang, Zhenzhen Xu

**Affiliations:** 1International Cooperation Research Center of Textile Structure Composite Materials, School of Textile & Garment, Anhui Polytechnic University, Wuhu 241000, China; xingjian@ahpu.edu.cn (J.X.);; 2Anhui Weiyi Textile Co., Ltd., Bozhou 236000, China

**Keywords:** polylactic acid, butenediol vinyl alcohol copolymer, composites, toughness, hydrophilicity

## Abstract

Due to its poor toughness and hydrophilicity, the application of polylactic acid (PLA) in the field of absorbent sanitary materials is restricted. A butenediol vinyl alcohol copolymer (BVOH) was used to improve PLA via melt blending. The morphology, molecular structure, crystallization, thermal stability, tensile property, and hydrophilicity of PLA/BVOH composites with different mass ratios were investigated. The results show that the PLA/BVOH composites possessed a two-phase structure with good interfacial adhesion. The BVOH could effectively blend into PLA without a chemical reaction. The addition of the BVOH promoted the crystallization of PLA, improved the perfection of the crystalline region, and increased the glass transition temperature and melting temperature of PLA in the heating process. Moreover, the thermal stability of PLA was markedly improved by adding the BVOH. The addition of the BVOH also had a significant effect on the tensile property of the PLA/BVOH composites. When the content of the BVOH was 5 wt.%, the elongation at the break of the PLA/BVOH composites could reach 9.06% (increased by 76.3%). In addition, the hydrophilicity of PLA was also significantly improved, and the water contact angles decreased with the increase in the BVOH content and time. When the content of the BVOH was 10 wt.%, the water contact angle could reach 37.3° at 60 s, suggesting good hydrophilicity.

## 1. Introduction

With the intensification of the aging population, the consumption of absorbent sanitary products for adults continues to grow. At present, the main raw material source of absorbent sanitary products is ES (ethylene propylene side by side) fiber, which is a polyolefin fiber derived from petroleum. The application of disposable absorbent sanitary products provides comfort, health, and convenience to people. However, the nonbiodegradability and non-renewability of ES fibers have led to serious white pollution due to the billions of products consumed annually [1,2]. In addition, the huge consumption has also intensified the depletion of oil resources [3,4,5]. Therefore, the replacement of traditional petroleum-based materials and the requirement for renewable carbon resources have become increasingly important and pressing [6,7,8].

To overcome these problems, researchers have developed new polymetric materials that can be biodegraded and obtained from renewable resources [9,10,11]. Among these biodegradable polymers, polylactic acid (PLA) has been widely considered a potential substitute to replace petroleum-based materials. As a renewable resource derived from natural biological materials, PLA possesses good degradability and biocompatibility, an appropriate mechanical property, and good processibility [12,13,14]. Therefore, PLA has been rapidly applied in industrial packing, hygienic products, apparel and household products, and other fields [15,16,17]. Moreover, PLA has also been hugely expected to be used in biomedical applications, such as drug delivery, blood vessels, tissue engineering, and scaffolding, due to its excellent biocompatibility, non-toxicity, and biodegradability [17,18].

Nonetheless, some inherent defects, such as poor toughness, low heat resistance, easy hydrolysis, a slow crystallization rate, and poor hydrophilicity, have limited the range of applications [18,19,20]. Modifications, such as copolymerization, plasticization, and blending with other polymers, have also been reported to improve the properties of PLA [18,21,22,23,24,25,26,27]. Among these modification methods, blending is considered to be the most cost-efficient and practical. Therefore, the research about PLA-based blends, including binary PLA blends, ternary polymer blends of PLA with two other polymers, and ternary blend nanocomposites of PLA with a polymer and nanofiller, has grown exponentially over the last two decades [18,22,24,27].

The absorbent sanitary products prepared by ES fibers need to be modified in a hydrophilic manner to meet comfort requirements. Therefore, the poor hydrophilicity of PLA seriously restricts its application in absorbent sanitary products, which also needs to be significantly improved. The hydrophilic modification of PLA can be mainly divided into blending modification and surface modification [18,28,29,30]. Surface modification is mainly divided into surface grafting and surface coating, which are the main hydrophilic modification methods for PLA fiber products. However, some drawbacks of surface modification, such as the complexity of the process, the use of an organic solvent, low fastness, and high cost, limit its application and need to be overcome.

It is well-known that melt blending modification is a handy, effective, and environmentally friendly method that can improve the hydrophilicity of PLA, but the difficulty is to screen out the suitable hydrophilic modifier, which can be well dispersed in a PLA matrix and have good compatibility with PLA. Poly (vinyl alcohol) (PVOH), which is a biodegradable, biocompatible, inexpensive, hydrophilic, and highly flexible polymer, has been used to improve the toughness and hydrophilicity of PLA by melt blending. The addition of PVOH can promote the hydrophilicity of PLA, but the immiscibility of PLA and PVOH degrades other properties [18].

A butenediol vinyl alcohol copolymer (BVOH) is composed of butenediol and vinyl alcohol monomers and is a new, multifunctional, environmentally friendly material that combines moldability and water-soluble biodegradability (Figure 1) [31,32,33]. It is important that BVOH exhibits good hydrophilicity due to the large number of hydroxyl groups in its molecular chains. Moreover, the good melt processability (a melting point of 180 °C) of a BVOH helps it improve the properties of other polymers via simple melt blending. Therefore, it is very appropriate to improve the hydrophilicity and toughness of PLA using a BVOH by melt blending due to its good biodegradability, mechanical property, similar melting point, and hydrophilicity. However, very few reports have examined the manufacturing and properties of PLA/BVOH composites [33]. In this work, the possibility of preparing PLA/BVOH composites via melt blending is explored. Moreover, the effect of the BVOH content on the morphology, crystallization behavior, thermal stability, tensile property, and hydrophilicity of the obtained PLA/BVOH composites is also discussed in detail.

## 2. Results and Discussion

### 2.1. Morphology of PLAB_x_ Composites

The morphologies of the fracture with liquid nitrogen of PLAB_x_ composites are shown in Figure 2. The dispersion of the BVOH in the PLA matrix and the interface state can be observed in Figure 2. It can be found that PLAB_x_ composites exhibit a clear two-phase structure (sea–island structure). The main component of PLA forms the continuous phase (sea phase), and the minor component of the BVOH forms the dispersed phase (island phase). It is evident that the BVOH can uniformly disperse in the PLA matrix with different contents in the particle state. In Figure 2a,b, it can be observed that the BVOH can be uniformly distributed in a PLA matrix in an irregular granular form with similar sizes when the content of the BVOH is equal to or less than 3 wt.%. Moreover, the size of the island phase ranges from 0.1 to 0.8 μm. With the increase in the BVOH content, the size of the island phase gradually increases, and the two-phase structure becomes more obvious. When the BVOH content is equal to or greater than 5 wt.%, the island phase mainly exists in a spherical form, and the size of the island phase ranges from 0.5 to 2.2 μm. The difference in island phase size also increases significantly when the BVOH content reaches 10 wt.%, as shown in Figure 2d.

Furthermore, it can be observed that there is no obvious boundary between the interface of the two phases when the BVOH content is equal to or less than 5 wt.%, as shown by the red scissors, indicating that the BVOH has good compatibility with the PLA component when the BVOH content is low [34]. This can be attributed to the close polarity due to the existence of a polar group, such as an alcoholic hydroxyl group in the BVOH and ester carbonyl in PLA. However, when the BVOH content reaches 10 wt.%, the appearance of the boundary between PLA and the BVOH can be observed, as shown by the blue scissors. This indicates that the compatibility of PLA and the BVOH begins to decline, which may be contributed to the change in surface tension caused by the morphological and dimensional variations in the BVOH phase.

Moreover, the void spaces throughout the fracture surface caused by the BVOH being pulled out of the PLA matrix can be observed in Figure 2. This phenomenon indicates that the interfacial adhesion between PLA and the BVOH is not significant [21,35]. With the increase in the BVOH content, the amount and size of the void spaces rapidly increase, especially when the BVOH content is up to 10 wt.%. This may be attributed to the excessive content and larger size of the BVOH. The bonding force between the interfaces of PLA and the BVOH decreases, and the island phase can be easily pulled out of the sea phase when subjected to an external force.

### 2.2. Chemical Structure of PLAB_x_ Composites

Figure 3 shows the FTIR spectrum of the PLA, BVOH, and PLAB_x_ composites. For pure PLA, the peak at 1748 cm^−1^ can be attributed to the stretching vibration of symmetric C=O. The peak at 1452 cm^−1^ belongs to the asymmetric bending deformation of -CH_3_. The peak at 1358 cm^−1^ can be attributed to the stretching vibration of -CH and asymmetric deformation of crystalline -CH_3_. The peaks at 1180 and 1081 cm^−1^ belong to the stretching vibration of C-O-C. The small peaks at 2996 and 2948 cm^−1^ can be assigned to the asymmetric and symmetric stretching vibration of -CH_3_, respectively. Moreover, a small band that can be attributed to the hydroxyl groups can also be detected at ~3358 cm^−1^ [21,36]. Meanwhile, the spectrum of the pure BVOH is characterized by an -OH stretching band at 3310 cm^−1^, -CH_2_ asymmetric and symmetric stretching bands at 2918 and 2851 cm^−1^, C-OH stretching bands at 1452 and 1088 cm^−1^, and a C-C stretching band at 1249 cm^−1^. Moreover, the small weak peak at 1652 cm^−1^ can be assigned to the terminal C=C stretching vibration. It needs to be pointed out that a peak at 1735 cm^−1^ can be observed in the FTIR spectrum of the BVOH, which can be attributed to the stretching vibration of C=O [18,33]. In Figure 1, it can be found there is no C=O in the macromolecular structure of the BVOH. Therefore, it can be inferred that the C=O comes from the byproducts of the BVOH synthesis.

In Figure 3, it can be observed that the main bands of PLA and the BVOH can also be found in the spectra of PLAB_x_ composites, and there are no absorption bands of new functional groups in the composites. Therefore, PLA and the BVOH physically blend without generating new chemical functional groups during the preparation process. Additionally, it can be found that the hydroxyl band of the PLAB_x_ composites increases in intensity with the increase in the BVOH content. This can be attributed to the contribution of hydroxyl groups in the BVOH. Moreover, it can be observed that the hydroxyl band of the PLAB_x_ composites shifts towards smaller wavenumbers by increasing the BVOH content, which indicates that hydrogen bonding occurs between the terminal hydroxyl groups of PLA and the hydroxyl groups of the BVOH [33,36].

### 2.3. Crystallization Behavior of PLAB_x_ Composites

The melting and crystallization curves of the pure PLA, BVOH, and PLAB_x_ composites are shown in Figure 4. The thermal parameters of the PLAB_x_ composites derived from DSC curves are listed in Table 1. Crystallinity (*X_c_*) is calculated based on the following equation:(1)$$X_{c}=\frac{{\Delta H}_{m}-{\Delta H}_{cc}}{{\Delta H}_{f}(1-x)},$$
where Δ*H_m_* is the melting enthalpy of the samples, Δ*H_cc_* is cold crystallization enthalpy, Δ*H_f_* is the melting enthalpy of 100% crystalline PLA (93 J/g) [36], and x is the content of the BVOH.

In Figure 4, it can be seen that the glass transition temperature (*T_g_*) in the reheating and cooling curves can both be detected, and the crystallization exothermic peak and melting endothermic peak are obvious, indicating that the BVOH is a crystalline polymer. In Figure 4b, it can be found that the crystallization exothermic peak of pure PLA appears as a slight bulge at about 102.7 °C in the cooling curves, implying a very low crystallization capacity. The crystallization enthalpy (Δ*H_c_*) is only 0.02 J/g, which also indicates that the vast majority of macromolecules in pure PLA are in the amorphous region. In the cooling curve of the BVOH, a glass transition temperature at 62.3 °C, and an obvious crystallization peak at 129.4 °C can be observed. Moreover, with the increase in the BVOH content, the crystallization exothermic peak becomes obvious, the temperature of the crystalline peaks (*T_c_*) shifts to low temperatures, and the Δ*H_c_* gradually increases. This phenomenon indicates that adding the BVOH can improve the crystallinity of PLA due to the high *T_c_* (129.4 °C) because the crystal grains of the BVOH can supply nucleation sites as a nucleating agent to promote the crystallization of PLA [37,38]. Moreover, the glass transition temperature in the cooling process (*T_gc_*) also moves towards low temperatures, which shows that the motility of PLA macromolecules is enhanced in the cooling process. When the content of the BVOH reaches 10 wt.%, two crystallization peaks at 97.9 and 124.2 °C can be observed from the cooling curves, which can be attributed to the PLA component and the BVOH component, respectively.

In Figure 4a and Table 1, it can be found that the pure PLA and PLAB_x_ composites exhibit glass transition, cold crystallization, and subsequent melting in the reheating process. The appearance of cold crystallization peaks indicates that incompletely crystallized chains exist in the PLA matrix. In the reheating curve of the BVOH, a glass transition temperature of 73.1 °C and an obvious melting peak at 179.8 °C can be observed. Moreover, with the addition of the BVOH, the glass transition temperature in the reheating process (*T_g_*_−_*_h_*) moves towards high temperatures. This indicates that the PLAB_x_ composites have a more thermally stable structure than pure PLA, which further exhibits that PLA and the BVOH are compatible. This phenomenon can be attributed to two factors. First, the crystallization of the BVOH in the cooling process can promote the crystallization of PLA, which can make the PLA macromolecular chain arrangement change from a disordered structure to an ordered structure. Therefore, it takes more heat to make the macromolecular chains move during the heating process. Second, the *T_g_* of the BVOH is much higher than that of PLA, so the BVOH molecular chains in the glassy state can hinder the movement of PLA macromolecular chains [8,21,38].

Furthermore, it can also be observed that the cold crystallization temperature (*T_cc_*) moves towards high temperatures and Δ*H_cc_* decreases with the addition of the BVOH. This can be explained by the fact that the movement restriction of the PLA macromolecular chains by the BVOH results in an increase in *T_cc_*. Moreover, the decrease in Δ*H_cc_* can be attributed to the larger crystallization region and a higher degree of crystal perfection in PLAB_x_ composites because of the promoting effect of the BVOH on the crystallization of PLA in the cooling process. Thus, pure PLA needs more crystal enthalpy to perfect the crystals than the PLAB_x_ composites. The increase in melting temperature (*T_m_*), Δ*H_m_*, and *X_c_* by adding the BVOH can also be crystalline and is explained by this reason. Moreover, it needs to be pointed out that there is only one melting peak in the reheating curves, which also illustrates the good compatibility of PLA and the BVOH.

### 2.4. Thermal Stability of PLAB_x_ Composites

The thermogravimetric analysis (TGA) and derivative thermogravimetric (DTG) curves of the BVOH, pure PLA, and PLAB_x_ composites are shown in Figure 5. The degradation temperature corresponding to a 5% mass loss (*T_5%_*), the maximum rate of decomposition (*T_max_*), and the end of decomposition (*T_end_*) are summarized in Table 2. In Figure 5 and Table 2, it can be found that the BVOH shows a three-stage thermal degradation at temperatures below 800 °C. The first stage of decomposition can be attributed to the volatilization of adsorbed water below 125 °C. This is because the BVOH macromolecules contain many hydrogen bonds that can combine adsorbed water. A noticeable finding is the second stage of decomposition (225–380 °C), which shows a high thermal decomposition peak. At this stage, the decomposition of the BVOH occurs. The final stage of decomposition (380–460 °C) further decomposes the BVOH due to the combustion of the pyrolysis residue [19,34]. Pure PLA shows a single-stage degradation within the measurement range, while PLAB_x_ composites exhibit a two-stage degradation. In Figure 5a and Table 2, it can be observed that the *T_5%_* of the PLAB_x_ composites is improved due to the addition of the BVOH, while the *T_5%_* of the PLAB_x_ composites decreases with the increase in BVOH content but is still higher than that of pure PLA. Moreover, the maximum rate of decomposition in the first stage (*T_max1_*) of PLA is also improved by adding the BVOH but decreases with the increases in the BVOH content. These phenomena indicate that adding the BVOH can improve the thermal stability of PLA. Moreover, the maximum rate of decomposition in the second stage (*T_max2_*) of the PLAB_x_ composites increases with the increase in the BVOH content.

### 2.5. Tensile Property of PLAB_x_ Composites

The stress–strain curves of the BVOH, pure PLA, and PLAB_x_ composites are shown in Figure 6a. The stress and elongation at the break of the PLAB_x_ composites are shown in Figure 6b. The stress yield phenomenon is not observed when pure PLA is stretched from Figure 5a, and the elongation at the break is only 5.23%, which exhibits brittle fractures. The elongation at the break of the BVOH can reach 22.9%, and the yield phenomenon is demonstrated. Moreover, the tensile property of PLA can be influenced by adding the BVOH. However, the PLAB_x_ composites still exhibit brittle fractures. The tensile stress of the PLAB_x_ composites first increases and then decreases with the increase in the BVOH content. The tensile stress of pure PLA is 53.4 MPa, and the tensile stress of PLAB_3_ composites can reach 65.9 MPa, with an increase of 23.4%. This can be attributed to the increase in crystallinity and the improvement in the perfection of the crystalline region due to the addition of the BVOH. The tensile stress begins to decrease when the BVOH content is higher than 5%. This may be due to the stress concentration caused by the increasing difference in the BVOH particle size. Moreover, the elongation at the break of the PLAB_x_ composites also first increases and then decreases with the increase in the BVOH content. The elongation at the break of pure PLA is only 5.23%, and the elongation at the break of PLAB_5_ composites can reach 9.06%, with an increase of 73.2%. This may be attributed to the good dispersion and compatibility of the BVOH in the PLA matrix.

### 2.6. Hydrophilicity of PLAB_x_ Composites

The water contact angles (WCAs) are used to characterize the hydrophilicity of the PLAB_x_ composites. In Figure 7, it can be observed that pure PLA exhibits poor hydrophilicity, and the WCA is 82.7° when the test droplets contact the samples. The WCAs of pure PLA decrease with the increase in time, and the WCA can reach 73.6° when the contact time is 60 s. This is because the terminal hydroxyl groups of PLA macromolecules exhibit hydrophilicity. Moreover, the WCAs of PLAB_x_ composites also decrease with the increase in the BVOH content. The WCA of the PLAB_10_ composite is 72.3° when the contact time is 0 s. This can be attributed to the good hydrophilicity of the BVOH due to a large number of hydroxyl groups. Furthermore, the WCAs of the PLAB_x_ composites also decrease with the increase in time, and the degree of decline is much greater than that of pure PLA. The WCA of the PLAB_10_ composite is only 37.3° when the contact time is 60 s, which indicates much better hydrophilicity than pure PLA. This can be explained by the fact that the large number of hydroxyl groups in the BVOH can make it dissolve in water. The water molecules can be diffused and wetted on the surface of the PLAB_x_ composites, and the WCA on the surface of the PLAB_x_ composites is reduced.

## 3. Materials and Methods

### 3.1. Materials

Polylactic acid (PLA) (Ingeo^TM^ Biopolymer 6100D, thermoplastic fiber-grade resin, with a specific gravity of 1.24 g/cm^3^) was supplied by Nature Works LLC (Plymouth, MN, USA). Butenediol vinyl alcohol copolymer (BVOH) (melting temperature 175 °C, melt volume flow rate 11.4 cm^3^/10 min, 210 °C, 2.16 kg ISO1133) was purchased from BASF (China) Co., Ltd. (Shanghai, China). All materials were dried in a vacuum oven before use to remove residual moisture and thus prevent hydrolysis during processing.

### 3.2. Sample Preparation

The PLA and BVOH pellets were first premixed in different mass ratios (100/0, 99/01, 97/03, 95/05, 90/10) and then melt blended using an SISZ-10A counter-rotating twin-screw extruder (Wuhan RuiMing Co., Ltd., Wuhan, China). The screw extruder adopted conical twin screws that were the meshing type and inward counter-rotating. The screw diameter was 10/25 mm from the small end diameter to the big end diameter. The screw length was 190 mm, and the screw length–diameter ratio was 19:1. The maximum screw speed could reach 51 rpm. Melt blending continued for 3 min with a screw speed of 30 rpm. The temperature from the feed zone to the nozzle was 185, 190, 190, and 190 °C, respectively. The PLA/BVOH composites are denoted herein as PLAB_x_, where x is the weight ratio of BVOH. Before melt blending, PLA resins were dried in a vacuum oven at 80 °C for 6 h, and BVOH pellets were dried at 60 °C for 8 h.

The specimens used for the tensile tests were manufactured by an SZS-20 injection-molding machine (Wuhan RuiMing Co. Ltd., China). The injection temperature was 200 °C, and the mold temperature was 65 °C. The pressure holding time was 5 s, and the cooling time was 30 s with an injection pressure of 25 MPa. Before the injection, the blends were dried in a vacuum oven at 80 °C for 12 h. Moreover, pure PLA pellets were also subjected to the same melt blending and injection treatment to obtain the same thermal history as the blends.

### 3.3. Characterization

The morphology of the composites was observed by a GeminiSEM 300 scanning electron microscope (SEM) (Carl Zeiss AG, Jena, Germany) at 3 kV with a cryo-fracture surface coating with a thin gold layer. A Nicolet-10 Fourier transform infrared spectrometer (FT-IR) (Thermo Fisher Scientific Co., Ltd., Waltham, MA, USA) was used to examine the chemical structures of the composites in the 4000–400 cm^−1^ range.

The thermal properties of the composites were determined using a DSC-3 differential scanning calorimetric (DSC) (Mettler Toledo Co., Ltd., Zurich, Switzerland). All samples were vacuum-dried at 80 °C for 12 h before the DSC test. The samples were subjected to a thermal cycle. In the first step, samples (5~8 mg) were heated from 25 to 210 °C with a heating rate of 10 °C/min and kept at that temperature for 3 min to eliminate the heat history. In the second step, the samples were examined from 30 to 210 °C with a heating and cooling rate of 20 °C/min. All tests were performed in a nitrogen atmosphere (50 mL/min). The thermal parameters for the crystallization and melting behavior of PLAB_x_ composites were evaluated from the second test step.

TGA/DSC 3+ thermogravimetric analyzer (TGA) (Mettler Toledo Co., Ltd., Zurich, Switzerland) was used to observe the thermal stability of the composites. The tests were performed in a nitrogen atmosphere (50 mL/min) at temperatures of up to 800 °C at a 10 °C/min heating rate.

Tensile properties of the composites were analyzed using an EZ-SX tensile tester (Shimadu Co., Ltd., Nagoya, Japan) with a 50 kN load cell according to the ASTM D638 with a loading speed of 5 mm/min. The testing of each sample was repeated at least five times, and the results were averaged

The water contact angles (WCAs) were measured using an OSA 60 video contact angle analyzer at 25 °C and 60% relative humidity. Deionized water was used as test liquid, and the volume of test drops was 5 μL. The average value was obtained by a multi-point test.

## 4. Conclusions

In this paper, PLAB_x_ composites composed of PLA and a BVOH were successfully prepared by simple melt blending using a twin-screw extruder. The PLAB_x_ composites exhibited a clear “sea-island” two-phase structure with good compatibility and interface adhesion between the PLA and BVOH phases when the BVOH content was low. The FTIR analysis shows that PLA and the BVOH experienced physical melt blending without generating new chemical functional groups. However, hydrogen bonding occurred between the hydroxyl groups of PLA and the BVOH. Moreover, the addition of the BVOH promoted the crystallization of PLA in the cooling process but hindered the movement of PLA macromolecular chains in the heating process. The thermal stability of PLA was also improved by adding the BVOH. The brittle fracture of PLA was not changed with the addition of the BVOH during the tensile test, but the elongation at the break still exhibited a significant increase. Furthermore, the hydrophilicity of PLA was significantly improved by adding the BVOH, and the hydrophilicity of PLA became better with the extension of time. As a consequence, the addition of the BVOH improved the toughness and hydrophilicity of PLA.

## Figures and Tables

**Figure 1 molecules-28-03627-f001:**
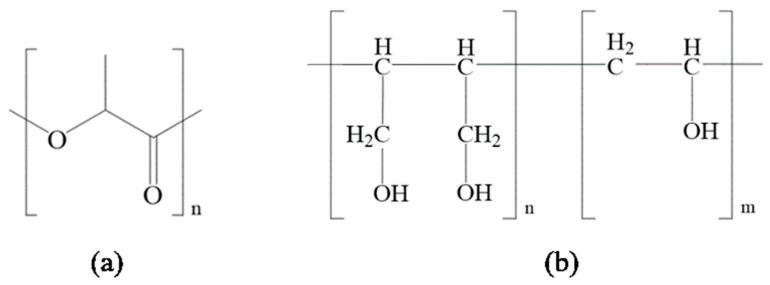
Chemical structure of (**a**) PLA and (**b**) BVOH.

**Figure 2 molecules-28-03627-f002:**
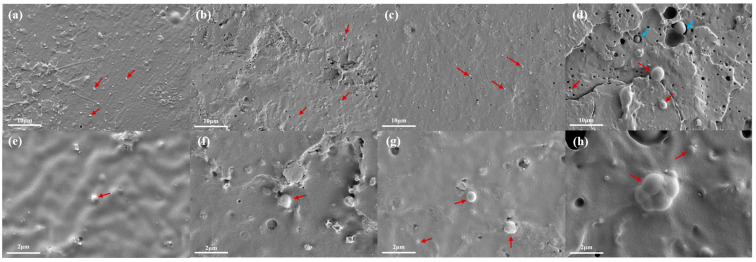
SEM images of PLAB_x_ composites: (**a**,**e**) 1 wt.%, (**b**,**f**) 3 wt.%, (**c**,**g**) 5 wt.%, (**d**,**h**) 10 wt.%. The red scissors represent good compatibility, and the blue scissors represent poor compatibility.

**Figure 3 molecules-28-03627-f003:**
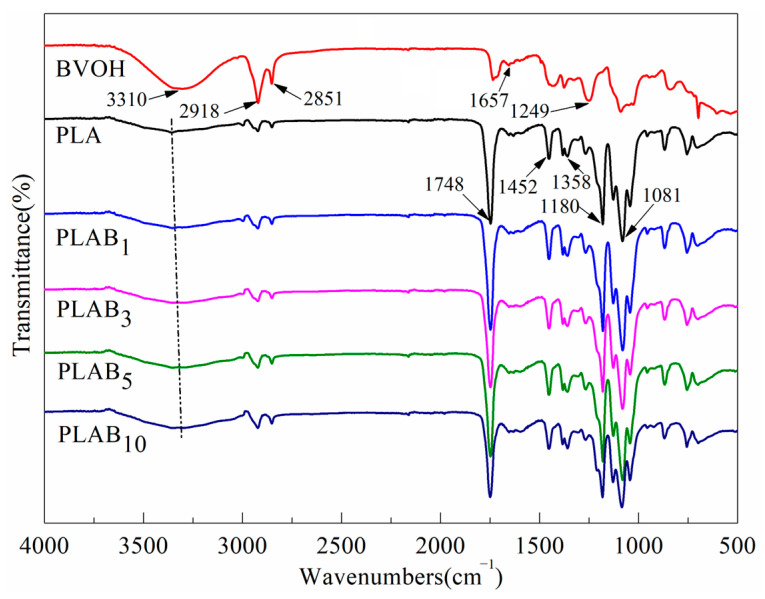
FTIR spectra of pure PLA, BVOH, and PLAB_x_ composites.

**Figure 4 molecules-28-03627-f004:**
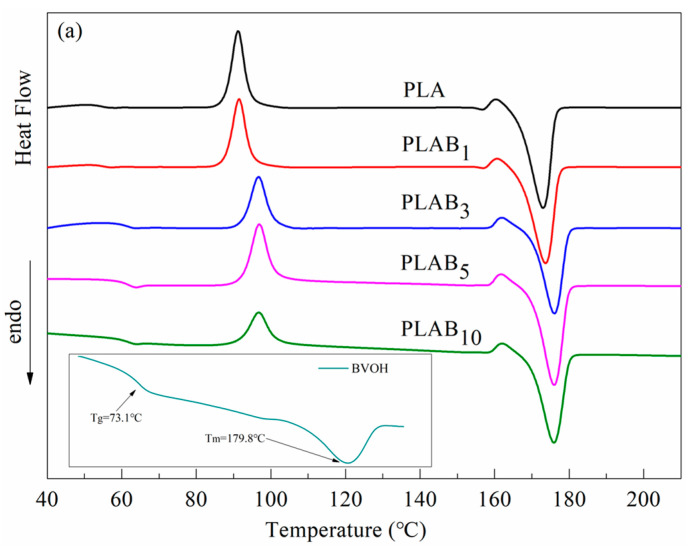
DSC curves of PLA, BVOH, and PLAB_x_ composites: (**a**) reheating curves; (**b**) cooling curves.

**Figure 5 molecules-28-03627-f005:**
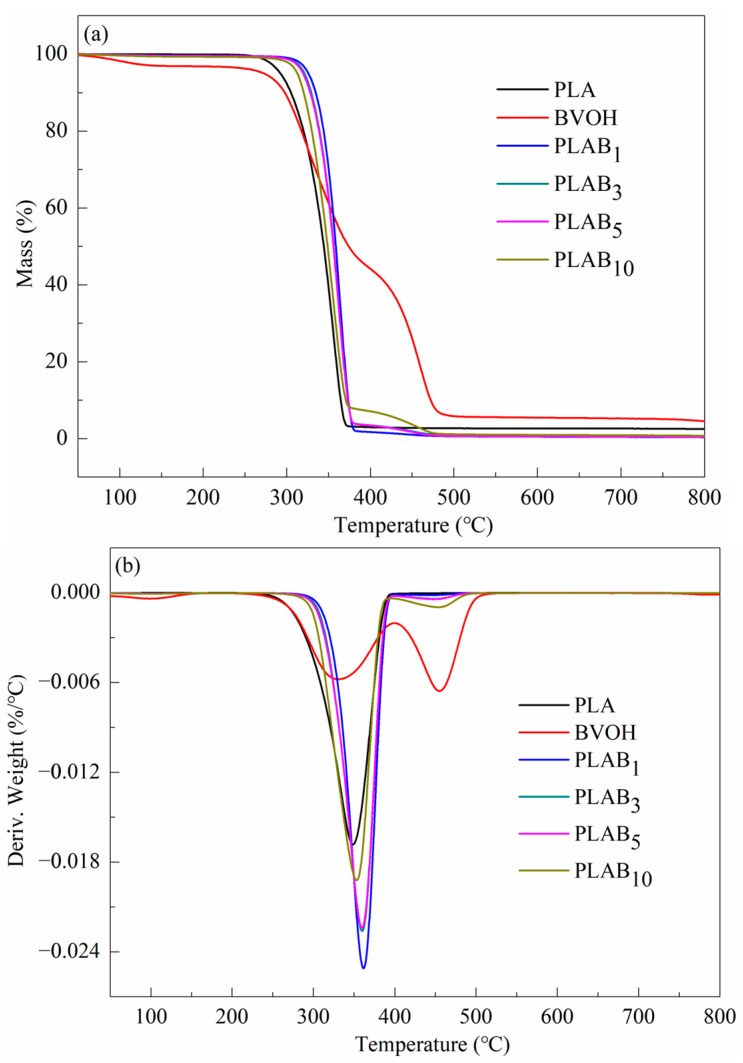
(**a**) TGA and (**b**) DTG curves of pure PLA, BVOH, and PLAB_x_ composites.

**Figure 6 molecules-28-03627-f006:**
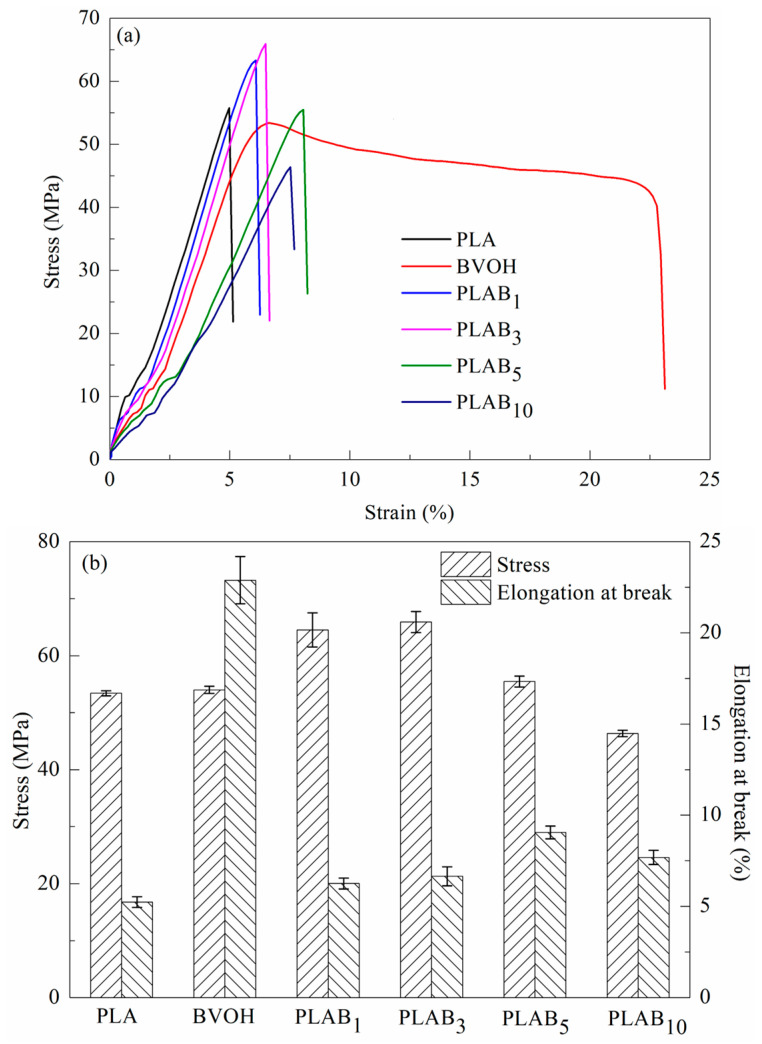
Tensile property of pure PLA, BVOH, and PLAB_x_ composites: (**a**) stress–strain curves, (**b**) stress and elongation at break with BVOH content.

**Figure 7 molecules-28-03627-f007:**
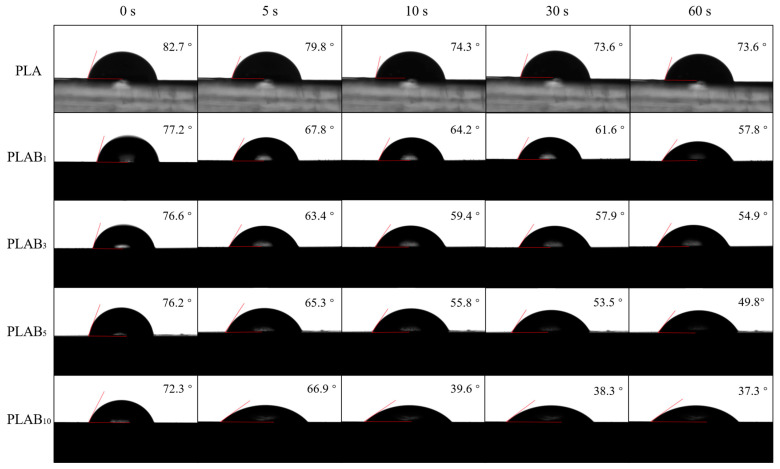
Hydrophilicity of pure PLA and PLAB_x_ composites at different contact times.

**Table 1 molecules-28-03627-t001:** Thermal parameters of BVOH, pure PLA, and PLAB_x_ composites.

Samples	*T_g_*_−*h*_ (°C)	*T_g_*_−*c*_ (°C)	*T_cc_* (°C)	Δ*H_cc_* (J/g)	*T_c_* (°C)	Δ*H_c_* (J/g)	*T_m_* (°C)	Δ*H_m_* (J/g)	*X_c_* (%)
PLA	53.8	57.1	91.4	5.03	102.7	0.02	173.1	14.71	10.41
PLAB_1_	54.7	51.4	91.9	4.68	97.3	0.73	173.9	15.04	11.25
PLAB_3_	59.7	50.3	96.7	3.99	97.3	0.64	176.2	16.81	14.43
PLAB_5_	60.3	50.2	97.3	3.78	97.5	0.66	176.5	24.81	23.58
PLAB_10_	61.2	54.2	97.6	3.15	97.9/124.2	1.70	176.5	28.35	30.11

**Table 2 molecules-28-03627-t002:** TGA parameters of BVOH, pure PLA, and PLAB_x_ composites.

Samples	BVOH	PLA	PLAB_1_	PLAB_3_	PLAB_5_	PLAB_10_
*T_5%_* (°C)	272.2	292.3	325.7	321.1	319.6	312.5
*T_max1_* (°C)	331.4	348.6	361.6	359.9	359.7	353.5
*T_max2_* (°C)	455.2	-	442.3	442.3	447.7	453.6
*T_end_* (°C)	501.7	376.5	395.7	469.9	482.4	489.7

## Data Availability

All the data in this work are contained within the article.

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
