# Peer review of "Morphology and Properties of Polylactic Acid Composites with Butenediol Vinyl Alcohol Copolymer Formed by Melt Blending"

_molecules, 2023, doi:10.3390/molecules28083627_

Round 1

Reviewer 1 Report

In this article, authors compounded the butenediol vinyl alcohol copolymer with polylactic acid (PLA) in different ratios and presented the characterization results. It’s been observed that a simple and proper study plan was designed and implemented. It has been emphasized in the article that in the field of absorbent sanitary materials is restricted, but the article does not provide a comparative perspective on the potential use of the produced compound in this field. Some points to be addressed by the authors are as follows:

-        The authors blended PLA and BVOH in a co-rotating twin-screw extruder. They have given information about the number of zones and zone temperatures, but there is no information about screw design. When blending polymers in twin screw extruders, screw design (forming elements consisted along the zones such as feed, kneading and pressurizing) is an important parameter for compounding so that authors better provide information on screw design.

-        It has been mentioned that PLA and BVOH are compatible because of no obvious boundary between the interface of two phases. But according to the SEM images it is evident that there is a gap between BVOH crystallites and the PLA matrix, and this statement can not be mentioned without concentration. It can be said for low concentrations.

-        Authors presenting and discussing the results of the compounding of two immiscible phases but there is no mention or discussion on compatibilizing  strategy.

-        The language should be thoroughly checked for minor errors.

Scale bar and legend of the SEM images are not easily read, better be emphasized.

Author Response

Response to the reviewer’s comments

Thank you for your comments. They are very important and give a constructive manner to improve the manuscript by more clearly explaining experimental methods and reword some expressions. We have carefully revised the manuscript according to your comments and advices. The following is a point-to-point response to your comments.

Reviewer 1:

The authors blended PLA and BVOH in a co-rotating twin-screw extruder. They have given information about the number of zones and zone temperatures, but there is no information about screw design. When blending polymers in twin screw extruders, screw design (forming elements consisted along the zones such as feed, kneading and pressurizing) is an important parameter for compounding so that authors better provide information on screw design.

Authors’ responses:

Thanks to your good comments. The screw design of the twin-screw extruder has been added in the revised manuscript. The screw extruder adopts conical twin screws which are meshing type and inward counter-rotating. The screw diameter is 10/25 mm from the small end diameter to the big end diameter. The screw length is 190mm, and the screw length-diameter ratio is 19:1. The maximum screw speed can reach 51 rpm. The corresponding revised portion is marked in red in the manuscript.

Reviewer 1:

It has been mentioned that PLA and BVOH are compatible because of no obvious boundary between the interface of two phases. But according to the SEM images it is evident that there is a gap between BVOH crystallites and the PLA matrix, and this statement cannot be mentioned without concentration. It can be said for low concentrations.

Authors’ responses:

Thanks to your good comments. We are sorry for our imprecise statement. From Figure 1, it can be found that there is no obvious boundary between the interface of PLA phase and BVOH phase when the BVOH content is equal or less than 5 wt.%, indicating that BVOH has a good compatibility with PLA component when the BVOH content is low. However, when the BVOH content reaches 10 wt.%, it can be observed the appearance of boundary between PLA and BVOH as shown by blue scissors. This indicates that the compatibility of PLA and BVOH begins to decline. The corresponding revised portion is marked in red in the manuscript.

Reviewer 1:

Authors presenting and discussing the results of the compounding of two immiscible phases but there is no mention or discussion on compatibilizing strategy.

Authors’ responses:

Thanks to your good comments. The discussion on compatibilizing strategy of PLA and BVOH had been added in the revised manuscript. When the BVOH content is low, the good compatibility of PLA and BVOH can be attributed to the close polarity due to the existence of polar group such as alcoholic hydroxyl group in BVOH and ester carbonyl in PLA. However, with the increase of BVOH content, the compatibility of PLA and BVOH decreases. This phenomenon may be contributed to the change of surface tension caused by the morphological and dimensional variation of BVOH phase. It needs to point out that the change of compatibility of PLA and BVOH is complex. Our analysis may be incomprehensive. The corresponding revised portion is marked in red in the manuscript.

Reviewer 1:

The language should be thoroughly checked for minor errors.

Authors’ responses:

Thanks to your good comments. A professional English editing service (MDPI Language Services) has also been employed to revise the manuscript. The detailed revisions and depictions are embodied by red font and can be found in the revised manuscript.

Reviewer 1:

Scale bar and legend of the SEM images are not easily read, better be emphasized.

Authors’ responses:

Thanks to your good comments. The scale bar and legend of the SEM images has been modified for clear observation. The revised figure is marked in red in the manuscript.

Reviewer 2 Report

A very good physicochemical work, deserving publication in Molecules. The subject is well chosen and the the physical techniques are well selected. The results are well explaned aned the conclusions are clear.

Author Response

Response to the reviewer’s comments

Thank you for your comments. They are very important and give a constructive manner to improve the manuscript by more clearly explaining experimental methods and reword some expressions. We have carefully revised the manuscript according to your comments and advices. The following is a point-to-point response to your comments.

Reviewer 2:

A very good physicochemical work, deserving publication in Molecules. The subject is well chosen and the physical techniques are well selected. The results are well explained and the conclusions are clear.

Authors’ responses:

Thank you for your recognition of our research work, and thank you again for your comments. Moreover, the manuscript has also been modified according to the comments of other reviewers.

Reviewer 3 Report

The work is devoted to the popular topic of modifying PLA plastic to improve its thermal and mechanical properties, as well as to increase its hydrophilicity. The work may not be called outstanding; it is rather a routine study. The work is free of serious drawbacks. It should, however, be noted a sufficiently large number of grammatical errors. Also, studies of direct relevance to the topic of this paper are not reflected in the Introduction, e.g:

https://doi.org/10.1016/j.ijbiomac.2018.12.002

https://doi.org/10.1016/j.ijbiomac.2023.123396

According to the Reviewer opinion, the IR spectra of the blends do not seem to contain BVOH signals.  In addition, the location of the double bond in the BVOH macromolecule (assigned to the band at 1735 cm-1) is unclear. It would be helpful to give the structure of BVOH in the article.

Author Response

Response to the reviewer’s comments

Thank you for your comments. They are very important and give a constructive manner to improve the manuscript by more clearly explaining experimental methods and reword some expressions. We have carefully revised the manuscript according to your comments and advices. The following is a point-to-point response to your comments.

Reviewer 3:

The work is devoted to the popular topic of modifying PLA plastic to improve its thermal and mechanical properties, as well as to increase its hydrophilicity. The work may not be called outstanding; it is rather a routine study. The work is free of serious drawbacks. It should, however, be noted a sufficiently large number of grammatical errors. Also, studies of direct relevance to the topic of this paper are not reflected in the Introduction, e.g:

https://doi.org/10.1016/j.ijbiomac.2018.12.002

https://doi.org/10.1016/j.ijbiomac.2023.123396

According to the Reviewer opinion, the IR spectra of the blends do not seem to contain BVOH signals. In addition, the location of the double bond in the BVOH macromolecule (assigned to the band at 1735 cm-1) is unclear. It would be helpful to give the structure of BVOH in the article.

Authors’ responses:

Thanks to your good comments. We are sorry for our grammatical errors in the manuscript. A professional English editing service (MDPI Language Services) has also been employed to revise the manuscript. The detailed revisions and depictions are embodied by red font and can be found in the revised manuscript.

Moreover, the studies of direct relevance to the topic of this paper have been cited in the revised manuscript. The corresponding revised portion is marked in red in the manuscript.

  1. Nofar, M.; Sacligil, D.; Carreau, P.J.; Kamal, M.R.; Heuzey M.C. Poly (lactic acid) blends: Processing, properties and applications. Int. J. Biol. Macromol. 2019, 125: 307-360.

33.Yu, D.Z.; Yang, Q.Q.; Zhou, X.X.; Guo, H.Y.; Li, D.W.; Li, H.X.; Deng, B.Y.; Liu, Q.S. Structure and properties of polylactic acid/butenediol vinyl alcohol copolymer blend fibers. Int. J. Biol. Macromol. 2023, 232: 123396.

We appreciate the reviewer’s good comments about the IR spectra. We took another close look at the IR spectra. In the IR spectra of BVOH, a small weak peak at 1652 cm-1 should be assigned to the terminal C=C stretching vibration. The peak at 1735 cm-1 can be observed from the FTIR spectrum of BVOH, which can be attributed to the stretching vibration of C=O. From Figure 1, it can be found there is no C=O in the macromolecular structure of BVOH. Therefore, it can be inferred that the C=O comes from the byproducts if BVOH synthesis. We appreciate the reviewer for pointing out the flaws in the manuscript again. The macromolecular structure of PLA and BVOH has been added in the revised manuscript.

Reviewer 4 Report

In this work, the authors describe the preparation composites composed of PLA and BVOH by a simple melt blending. I am happy to see that the authors have read the most recent literature about the field because the references are in the most of cases in the last years. I consider the work interesting and appropriate for publication. However, before publication some little aspects should be improved:

1.      Every person who knows about chemistry must know what the chemical structure of PLA is or BVOH. However, it would be clearer if you write the chemical structure to clarify to the reader about the structures mentioned in the text.

2.      In Figure 1, it would be possible to improve the quality of the images in terms of the scales? They are difficult to see.

3.      Do you know that the contact angles for PLA will keep the same with the time? Do you know that the contact angle will be 73.6º at 90, 120, 150 s, etc…?

Author Response

Response to the reviewer’s comments

Thank you for your comments. They are very important and give a constructive manner to improve the manuscript by more clearly explaining experimental methods and reword some expressions. We have carefully revised the manuscript according to your comments and advices. The following is a point-to-point response to your comments.

Reviewer 3:

  1. Every person who knows about chemistry must know what the chemical structure of PLA is or BVOH. However, it would be clearer if you write the chemical structure to clarify to the reader about the structures mentioned in the text.

Authors’ responses:

Thanks to your good comments. The macromolecular structure of PLA and BVOH has been added in the revised manuscript.

Reviewer 3:

  1. In Figure 1, it would be possible to improve the quality of the images in terms of the scales? They are difficult to see.

Authors’ responses:

Thanks to your good comments. We are sorry for the unclear scales in the Figure 1. The scale bar and legend of the SEM images has been modified for clear observation. The revised figure is marked in red in the manuscript.

Reviewer 3:

3.Do you know that the contact angles for PLA will keep the same with the time? Do you know that the contact angle will be 73.6º at 90, 120, 150 s, etc…?

Authors’ responses:

Thanks to your good comments. In our research, the measure time of dynamic contact angles was 2 min. We found that there is no obvious change of contact angles after 60s, thus we set the test time to 60s in our manuscript. The contact angles for PLABx composites can significantly change with time within 60s, and the contact angles have no significant change with time above 60s.

Round 2

Reviewer 1 Report

The responses to the review provided by the authors and the proper modifications on the manuscript are found to be satisfactory. The article was found to be publishable in its present form with minor langıage revisions.